# Elevated Thyroxine Concentration and Lithium Intoxication—An Analysis Based on the LiSIE Retrospective Cohort Study

**DOI:** 10.3390/jcm11113041

**Published:** 2022-05-27

**Authors:** Ingrid Lieber, Michael Ott, Robert Lundqvist, Mats Eliasson, Mikael Sandlund, Ursula Werneke

**Affiliations:** 1Sunderby Research Unit, Department of Clinical Sciences, Psychiatry, Umeå University, 90185 Umeå, Sweden; ursula.werneke@umu.se; 2Department of Public Health and Clinical Medicine, Umeå University, 90185 Umeå, Sweden; michael.ott@umu.se; 3Sunderby Research Unit, Department of Public Health and Clinical Medicine, Umeå University, 90185 Umeå, Sweden; robert.lundqvist@norrbotten.se (R.L.); mats.eliasson@norrbotten.se (M.E.); 4Department of Clinical Sciences, Division of Psychiatry, Umeå University, 90185 Umeå, Sweden; mikael.sandlund@umu.se

**Keywords:** hyperthyroxinaemia, hyperthyroidism, thyroxine, lithium, intoxication, bipolar disorder, schizoaffective disorder, thyroid disorder

## Abstract

(1) Background: It has been suggested that hyperthyroxinaemia is a risk factor for lithium intoxication by altering tubular renal function. (2) Methods: We determined the relevance of hyperthyroxinaemia as a risk factor for lithium intoxication in patients with bipolar or schizoaffective disorder in the framework of the LiSIE (Lithium-Study into Effects and Side Effects) retrospective cohort study. Of 1562 patients included in the study, 897 patients had been exposed to lithium at any time between 1997 and 2017 with 6684 person-years of observation. (3) Results: There were 65 episodes of unintentional lithium intoxication in 53 patients. There were nine episodes with hyperthyroxinaemia at the time of lithium intoxication, yielding an incidence of 1.3 episodes/1000 person-years. For all nine episodes, we could identify alternative, more plausible, explanations for the observed lithium intoxications. (4) Conclusions: We conclude that hyperthyroxinaemia-associated unintentional lithium intoxication is an uncommon event. A direct causal link between hyperthyroxinaemia and altered tubular renal function remains elusive. Increasing the frequency of routine thyroid function tests seems unlikely to decrease the risk of lithium intoxication.

## 1. Introduction

Lithium-associated hyperthyroidism is 4–15 times rarer than lithium-associated hypothyroidism [1,2,3]. Yet it may be of substantial clinical significance, being able to destabilise mood and mimic manic episodes [2]. Lithium-associated hyperthyroidism may also affect somatic health. Concerns have been raised that thyroxine, through its effect on tubular function, may alter the renal clearance of lithium, thereby increasing the risk of lithium toxicity [4]. However, the evidence remains anecdotal; we could only find a few cases of lithium intoxication associated with elevated thyroxine concentrations related to thyrotoxicosis reported in the literature [4,5,6,7,8,9]. Therefore, the question arises of whether elevated thyroxine concentrations (hyperthyroxinaemia) could be a clinically relevant risk factor for lithium intoxication. If yes, patients at risk of or already affected by hyperthyroidism or patients experiencing transient thyroxine elevations, for instance in the context of an acute somatic or psychiatric illness [10,11], might need tighter monitoring of lithium serum concentrations.

### Aims

We sought to determine the relevance of hyperthyroxinaemia as a risk factor for lithium intoxication. Specifically, we tested the following hypotheses:Hyperthyroxinaemia is commonly associated with lithium intoxication.Hyperthyroxinaemia leads to increased tubular reabsorption of lithium, which increases the risk of lithium intoxication.

## 2. Materials and Methods

### 2.1. Study Design

This study was conducted in the framework of the LiSIE (Lithium—Study into Effects and Side Effects) research programme. LiSIE is a retrospective cohort study, based on a review of medical records, aiming at identifying the best long-term treatment options for patients with bipolar disorder and related conditions. LiSIE explores the effects and potential adverse effects of lithium compared to other mood stabilisers. The study was conducted according to the guidelines of the Declaration of Helsinki and approved by the Regional Ethics Review Board at Umeå University, Sweden (DNR 2010-227-31M, DNR 2011-228-32M, DNR 2014-10-32M, DNR 2018-76-32M). 

### 2.2. Lithium—Study into Effects and Side Effects Participants

LiSIE invited all individuals in the Swedish regions of Västerbotten and Norrbotten ≥18 years of age, who had (a) received a diagnosis of bipolar disorder (BD) (ICD10 F31) or schizoaffective disorder (SZD) (ICD10 F25), or (b) used lithium as a mood stabiliser between 1997 and 2011 [12]. All participants were informed about the nature of the study in writing and provided verbal informed consent. The consent was documented in our research files, dated, and signed by the research worker who obtained the consent. In accordance with the ethics approval granted, we also included deceased patients. Consent procedures concluded by the end of 2012. Thereafter, the cohort was locked, and no new patients were included in the study.

### 2.3. Patient Selection and Inclusion Criteria

For the current study, we considered patients from the region of Norrbotten who had received a diagnosis of either BD or SZD on at least two occasions, at least six months apart at any time between 1997 and 2013. We then selected all patients who had been treated with lithium at some time during a 21-year review period from 1997 to 2017. For patients with lithium exposure, we identified patients who had experienced at least one episode of lithium intoxication (Figure 1).

### 2.4. Exclusion Criteria

For the whole LiSIE study, we excluded patients in whom, after manual validation from the medical records, a diagnosis of schizophrenia was more likely than BD or SZD. We excluded all episodes of intentional intoxications. These could not be considered adverse drug reactions. We then excluded patients who did not have any free thyroxine (fT4) measurements available within three months before the event of a lithium intoxication.

### 2.5. Outcome Definition

The outcome of this study was episodes of unintentional lithium intoxication. Here, we considered only clinically relevant intoxications with a lithium serum concentration (s-lithium) of at least 1.5 mmol/L [13,14].

Grading the frequency of lithium intoxication as an adverse drug reaction.

We graded the occurrence of lithium intoxications according to Council for International Organizations in Medical Sciences (CIOMS) guidelines [15] (Table 1).

### 2.6. Exposure Parameters

#### 2.6.1. Lithium Exposure

Proof of lithium exposure was determined by at least one blood lithium concentration >0.2 mmol/L. As we investigated a potential adverse effect of lithium treatment but not the therapeutic effect, we did not require lithium concentrations to be therapeutic. For each patient, we validated the lithium start and stop date using available lithium concentrations, electronic prescriptions, and information from the medical records. Observed time in the study was measured from 1st of January 1997 to 31st of December 2017. The observation time for each individual patient started at the date of lithium initiation. For patients who moved out of the region or died before the 31st of December 2017, the observation time stopped at the date of their departure or death. We estimated the time of lithium exposure in person-years.

#### 2.6.2. Hyperthyroxinaemia

We considered patients to have experienced an episode of hyperthyroxinaemia if they had fT4 above the upper reference range. We only considered episodes of hyperthyroxinaemia that had occurred after the initiation of lithium. Most tests were analysed with an immunoassay from Roche Diagnostics Scandinavia with normal range reference values for thyroid function tests of 12.0–22.0 pmol/L for fT4. Hyperthyroxinaemia at the time of lithium intoxication was determined by fT4 tests taken within three months prior to the intoxication. If several tests were available, we chose the test closest before the lithium intoxication.

#### 2.6.3. Renal Function

We explored renal function before and during lithium intoxication. Creatinine in serum samples had been measured using an assay traceable to isotope dilution mass spectrometry (IDMS) creatinine. From creatinine, we calculated the estimated glomerular filtration rate (eGFR) using the The Chronic Kidney Disease Epidemiology Collaboration (CKD-EPI) 2021 formula [16].

#### 2.6.4. Other Variables

We also recorded age, sex, and type of underlying disorder. For episodes of lithium intoxication, we explored co-medications at the time of lithium intoxication that might have interfered with the renal clearance of lithium.

### 2.7. Chart Review, Analysis, and Validation

For the outcomes and exposure variables, we retrospectively reviewed the medical records of all eligible patients from 1997 up to 31 December 2017. From the medical records, we manually validated the date of the electronic prescriptions when lithium had been started or discontinued. 

### 2.8. Control for Bias and Missing Data

We controlled for selection bias in the whole retrospective cohort study (LiSIE) using key parameters available in anonymised form. These included age, sex, and where applicable, maximum recorded lithium and creatinine concentrations. In accordance with the ethics approval granted, we compared these parameters for consenting and non-consenting patients. No significant difference was found between the two groups. The data were complete for the defined outcome because lithium intoxication tended to be well documented and followed up. For one episode, a pre-intoxication fT4 measurement was not available.

### 2.9. Statistical Analysis

For hypothesis 1, we calculated the observed incidence of hyperthyroxinaemia-associated unintentional lithium intoxication based on identified episodes. We expressed the incidence in episodes/person-years. For hypothesis 2, we qualitatively explored the observed episodes of hyperthyroxinaemia-associated unintentional lithium intoxication. We examined if any unintentional lithium intoxication identified might be explained by increased tubular reabsorption attributable to hyperthyroxinaemia, or whether there was an alternative, more plausible, explanation. This could be GFR impairment due to dehydration or comedication with medicines affecting renal function. All data were anonymised before analysis. The analysis was conducted with SPSS version 27.0 (IBM, Armonk, NY, USA). The method adhered to the STROBE (Strengthening the Reporting of Observational studies in Epidemiology) checklist (Appendix A).

## 3. Results

### 3.1. Baseline Characteristics

A total of 1562 patients were included in the study, 62% women. Of these, 897 patients had been exposed to lithium at any time during the review between 1997 and 2017 with 6684 person-years of observation (Table 2).

Overall, there were 91 episodes of lithium intoxication in 74 patients (68.1% women). The incidence was 13.6 episodes/1000 person-years. There were 65 episodes of unintentional lithium intoxication in 53 patients (64.2% women). The incidence was 9.7/1000 person-years. In terms of CIOMS guidelines, this was an uncommon event. Whilst exposed to lithium, 138 (15.4%) patients experienced 260 episodes of hyperthyroxinaemia. The incidence was 38.9 episodes/1000 person-years. In terms of CIOMS guidelines, this was a common event.

### 3.2. Hypothesis 1: Hyperthyroxinaemia Is Commonly Associated with Lithium Intoxication 

There were 51 episodes in 41 patients of unintentional lithium intoxication with fT4 available at the time of intoxication. In these, there were nine episodes (cases) in whom hyperthyroxinaemia was diagnosed at the time of lithium intoxication, yielding an incidence of 1.3 episodes/1000 person-years. In terms of the CIOMS guidelines, this was an uncommon event.

### 3.3. Hypothesis 2: Hyperthyroxinaemia Leads to Increased Tubular Reabsorption of Lithium, Which Increases the Risk of Lithium Intoxication

Our second hypothesis held that hyperthyroxinaemia altered tubular function in the kidney, which then increased the risk of lithium intoxication. In all nine identified episodes, eGFR was decreased at the time of lithium intoxication. In six episodes (episodes 1, 3, 4, 6, 8, and 9), eGFR was reduced by more than 50% compared to baseline. In three episodes (episodes 1, 5, and 8), diuretics had been added shortly before the intoxication. Additionally, in two of these episodes (episodes 1 and 5), medication affecting the renin-angiotensin-system had been started. In episodes 6 and 7, reduced eGFR could be explained by dehydration. In episode 2, the aetiology of reduced eGFR remains unclear. In seven episodes (cases 3–9), fT4 did not exceed 30pmol/L. In only three episodes, the peak s-lithium concentration exceeded 2.0 mmol/L (Table 3).

## 4. Discussion

In our study, observed hyperthyroxinaemia-associated unintentional lithium intoxication was an uncommon event. Therefore, we judge hypothesis 1 to be incorrect. According to hypothesis 2, hyperthyroxinaemia led to increased tubular reabsorption of lithium, increasing the risk of lithium intoxication. In all nine episodes with elevated fT4 at the time of lithium intoxication, eGFR was reduced at the time of lithium intoxication. As lithium is exclusively eliminated by glomerular filtration, reduced eGFR is a more plausible explanation for the observed lithium intoxications than increased tubular reabsorption. In a state of volume depletion, e.g., dehydration or as a consequence of diuretic treatment, lithium reabsorption in the proximal tubule is increased [17]. However, for each case, we could identify an alternative plausible mechanism with more potential to alter lithium concentration. Therefore, it seems unlikely that fT4-induced increase in tubular reabsorption played a role in the aetiology of lithium intoxication in our study, and we could not confirm hypothesis 2.

Very few studies have explored the incidence of lithium intoxication. In our current study, we expanded on previous work based on the same database [13]. In this study including even depressive patients on lithium, 7.2% of patients treated with lithium had experienced an episode of lithium intoxication ≥1.5 mmol/L. This yielded an incidence of eight episodes per 1000 person-years. Data extrapolated from US 2019 prescription data and US Poison control centres data suggests a higher annual prevalence of lithium intoxications between 0.9% and 1.6%, depending on whether lithium was mentioned as a single substance (Single Exposures) or as one of several substances (Case Mentions) The higher prevalence may arise from all intoxications, and not only intoxications ≥1.5 mmol/L, being counted in the US data. In fact, considering all episodes with lithium concentrations ≥1.2 mmol/L in our previous study would have yielded an annual prevalence of 1.7% [13].

Hyperthyroidism caused by lithium is considered rare. Estimates of incidence rates range from 0.08–4.8/1000 person-years [1,3,18,19]. However, hyperthyroxinaemia seems more common, and abnormal thyroid function tests can occur in up to one out of every three psychiatric patients without reflecting actual thyroid disease [20]. Thyroxine concentrations higher than expected without TSH suppression (euthyroid hyperthyroxaenemia) can occur in a variety of somatic conditions [21]. Several pharmacological substances can also increase thyroxine concentrations [22]. Acute psychiatric disorders have also been linked to elevated thyroxine concentrations for reasons yet to be understood [11,23]. Frequent testing of thyroid function may pick up cases of transient thyroxine elevations [7]. Such might otherwise go undetected if symptoms were only mild. 

The theory that thyroxine can alter the renal clearance of lithium leading to increased tubular reabsorption [4] stems from animal experiments. In rat experiments, it has been shown that hyperthyroidism may affect the activity of renal Na^+^-H^+^ exchange. This may then increase proximal tubular reabsorption of lithium and hence increase the risk of lithium intoxication [24]. A study in humans showed that the fractional excretion of lithium was significantly lower in hyperthyroid patients in comparison with euthyroid control subjects. The fractional excretion of lithium correlated to thyroxine and normalised with antithyroid treatment [25]. Our study concerned the impact of hyperthyroxinaemia as a risk factor for lithium in-toxication irrespective of the reason for the hyperthyroxinaemia. For all nine episodes with elevated fT4 at the time of lithium intoxication, we could identify an alternative more plausible explanation for the lithium intoxication. Therefore, we did not explore the length of exposure to hyperthyroxinaemia any further, even it can vary depending on the cause of the hyperthyroxinaemia.

On the other hand, hyperthyroxinaemia may increase glomerular filtration [26,27]. This should increase the elimination of lithium, thereby decreasing the risk of lithium intoxication. Here, thyroid hormones may directly affect kidney function through changes in renin-angiotensin-system activity, ion channels and transporters of the kidney. This can lead to upregulation of the Na^+^-K^+^-2Cl^−^ cotransporter and sodium and lithium uptake. Thyroid hormones may also indirectly affect kidney function through systemic effects. These include (a) decrease in systemic vascular resistance by enhanced nitric oxide synthase activity, (b) increased cardiac output by enhanced inotropy and chronotropy, and (c) increased renal plasma flow and blood pressure [26,28,29,30,31]. Therefore, thyroxine may have opposite effects on kidney function through tubular and glomerular mechanisms. Whether thyroxine can increase or decrease the risk of lithium intoxication may depend on the net effects of these opposing tubular and glomerular mechanisms.

Other symptoms of elevated thyroxine concentrations, such as severe diarrhoea, dehydration, and acute decompensated heart failure, may decrease GFR and lower excretion of lithium [6,25]. Elevated thyroid hormones may also lead to impaired cognition [32], which then could increase the risk of accidental overdoses. This would constitute an alternative link between hyperthyroidism and lithium intoxication, independent of renal failure or dehydration. Such cases may be difficult to recognise because lithium intoxication also changes cognition. Finally, it is possible that thyroxine concentrations raise as a consequence of the acute illness arising from lithium intoxication. However, this remains a hypothesis to be explored. Ultimately, symptoms of hyperthyroidism and lithium intoxication overlap. Thyroid function testss may be warranted if clinicians suspect that elevated lithium concentrations cannot fully account for the clinical presentation or symptoms do not resolve with lithium elimination.

We found six published cases that report an association between elevated thyroxine concentrations and lithium intoxication. In cases 1–4 [4,5,6,7], lithium was used as a treatment for an underlying affective disorder. In cases 5 and 6 [8,9], lithium was tested as a treatment for hyperthyroidism. In cases 1 and 2, eGFR was substantially reduced. In these cases, it was unlikely that elevated fT4 had caused the lithium intoxications through increased tubular reabsorption. In cases 3, 5 and 6 [5,8,9], the glomerular function was normal, so increased tubular reabsorption of lithium was a possible reason. In case 4 [7], there was not enough information to establish the cause of lithium intoxication (Table 4). 

### 4.1. Strengths

For this study, we had available 21 years of real-life validated data for 1562 patients, covering 81% of patients with BP/SZD in the region of Norrbotten, Sweden. Access to laboratory data, prescription data and medical records made it possible to establish the time treated on lithium so that we could calculate incidence rates in person-years. From the data sources available, we could also map the chronology of events to determine what came first, hyperthyroxinaemia or lithium treatment. The access to detailed clinical data made it possible to address our hypotheses with much greater complexity than possible in longitudinal register studies.

### 4.2. Weaknesses

The study was observational and retrospective in nature. This limited our ability to make inferences about causality. The study relied on retrospectively collated information from medical records. Hence, the quality of our results depended on the quality of the documented clinical information. However, we systematically abstracted the clinical information from three sources, laboratory data, prescription data and clinical notes. This way, we reduced the potential for misclassification beyond what is possible in observational studies based on registered data. Due to the rarity of the outcome, the study did not yield itself to a prospective design. FT4 measurements were not always taken right at the time of lithium intoxication. This could lead to a potential underestimation of the association between hyperthyroxinaemia and lithium intoxication. We could not judge any effect on tubular function directly because fractional lithium excretion was not available for any of the episodes. Instead, we had to make a clinical assessment in terms of clinical plausibility, taking into account other available information, such as eGFR, dehydration, and other medications altering kidney function. In our previous work on lithium intoxication based on the LiSIE study, 4.4% of episodes were linked to treatment with non-steroidal anti-inflammatory drugs (NSAID), 7.7% to blockade of the renin-angiotensin-aldosterone-system (RAAS) with angiotensin-converting enzyme inhibitors (ACEI), angiotensin receptor blockers or spironolactone and 3.3% to the use of thiazide or loop diuretics [13].

## 5. Conclusions

Hyperthyroxinaemia-associated unintentional lithium intoxication seems an uncommon event. A direct causal link between hyperthyroxinaemia and lithium intoxication via altered tubular renal function remains elusive. Clinicians should remain vigilant regarding common risk factors for lithium intoxication, such as dehydration or the use of medications that can change renal function. Based on the findings of our study, increasing the frequency of routine thyroid function tests seems unlikely to decrease the risk of lithium intoxication.

## Figures and Tables

**Figure 1 jcm-11-03041-f001:**
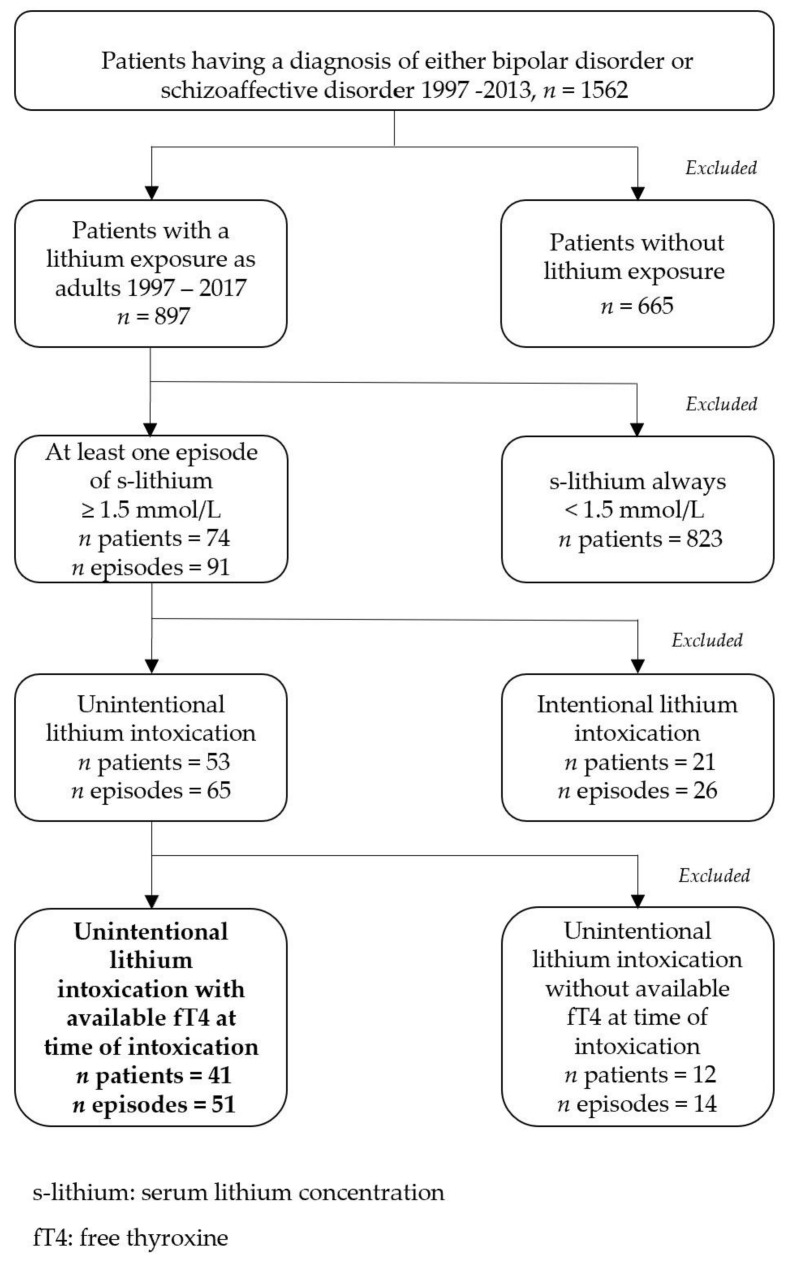
Selection of study sample.

**Table 1 jcm-11-03041-t001:** Frequency of adverse drug reactions according to Council for International Organizations in Medical Sciences (CIOMS).

Very Common	≥1/10	≥10%
Common (Frequent)	≥1/100 and <1/10	≥1% and <10%
Uncommon (Infrequent)	≥1/1000 and <1/100	≥0.1% and <1%
Rare	≥1/10,000 and <1/1000	≥0.01% and <0.1%
Very Rare	<1/10,000	<0.01%

**Table 2 jcm-11-03041-t002:** Baseline characteristic of sample.

Patients Exposed to Lithium, *n*	897
Sex, *n* (%) Male Female	357 (39.8) 540 (60.2)
Age (years) at study start Mean (SD) Median (min–max)	45.0 (15.2) 45.0 (18–92)
Type of diagnosis, *n* (%) Bipolar disorder Schizoaffective disorder	768 (85.6) 129 (14.4)
Time of lithium exposure (person–years) Total Mean (SD) Median (min–max)	6684 7.5 (6.5) 5.7 (0–21)

Max: maximum; min: minimum; n: number; SD: standard deviation.

**Table 3 jcm-11-03041-t003:** Presumed cause of lithium intoxication.

Episode ^a^	fT4 (pmol/L) ^b^		fT3 (pmol/L) ^c^		*eGFR* (mL/min/1.73m^2^)		s-Lithium (mmol/L)		NDI	Presumed Main Cause of Lithium Intoxication	
	Before	During	Before	During	Before	During	Before	During		↑ tubular reabsorption attributable to ↑ fT4	Alternative explanation:
1	22.3	84.1	NA	17.3	51	22	0.66	1.61	No	Unlikely	Addition of ARB and spironolactone 4 weeks before intoxication leading to ↓ GFR
2	16.1	30.1	3.1	5.7	85	64	0.82	1.74	No	Unlikely	↓ GFR, reason unclear
3	19.3	28.1	3.9	NA	67	15	0.81	4.20	No	Unlikely	Infection/pyelonephritis leading to ↓ GFR
4	19.5	24.7	NA	NA	37	6	0.83	1.50	No	Unlikely	Postrenal AKI leading to ↓ GFR
5	NA	24.7	NA	NA	64	42	0.62	2.59	No	Unlikely	ACEI and thiazide 11 weeks before leading to ↓ GFR
6	18.9	24.1	3.9	2.7	89	41	0.67	1.57	No	Unlikely	Dehydration and colitis leading to ↓ GFR
7	14.8	23.4	3.5	2.0	72	62	0.62	1.56	Yes	Unlikely	NDI and dehydration leading to ↓ GFR
8	13.7	22.6	3.4	4.9	54	26	0.69	2.02	No	Unlikely	Treatment with amiloride/hydrochlorothiazide 2 weeks before leading to ↓ GFR
9	20.8	22.4	5.2	NA	46	5	0.31	1.81	No	Unlikely	Sepsis and prerenal AKI leading to ↓ GFR

ACEI: angiotensin-converting enzyme inhibitor; AKI: acute kidney injury; ARB: angiotensin receptor blockers; eGFR: estimated glomerular filtration rate based on creatinine; fT3: free triiodothyronine; fT4: free thyroxine; NA: not available; NDI: nephrogenic diabetes insipidus diagnosed during or before lithium intoxication; s-lithium: serum lithium concentration. ^a^ Each episode occurred in a different patient. ^b^ Upper normal limit for fT4: 22 pmol/L. ^c^ Upper normal limit for fT3: 6.8 pmol/L.

**Table 4 jcm-11-03041-t004:** Published cases reporting an association between elevated thyroxine concentrations and lithium intoxication.

Case	Study	Sex	Age (years)	Li_max_ mmol/L	fT4_intox_ pmol/L (Upper Normal Reference)	Creatinine µmol/L (Upper Normal Reference) eGFR ^a^	Presumed Cause of Lithium Intoxication ^b^	
							Elevation fT4 Mediated (Dehydration/Tubular)	Alternative Explanation
**Lithium Used as Treatment for an Affective Disorder**								
**1**	[4]	F	34	3.27	72.4 (25.0)	387 (100) 13	Unlikely	Nephrogenic diabetes insipidus leading to dehydration leading to AKI and ↓ GFR Thioridazine interaction with lithium. Hyperthyroidism mediated dehydration possible
**2**	[6]	F	64	3.81	57.3 (19.1)	226 (71) 20	Unlikely	Hyperthyroidism mediated dehydration possible AKI due to another reason leading to ↓ GFR cannot be excluded
**3**	[5]	F	46	3.6	38.6 (23.2)	Normal	Possible	No
**4**	[7]	F	36	1.6	FT4I 21.6 (4–12ng/L) ^c^	Not known	Not enough information to rate	
Lithium Used as a Treatment for Hyperthyroidism								
**5**	[8]	F	37	3.40	103 (23.2)	Normal	Possible	No
**6**	[9]	F	66	1.54	31.8 (21.9)	Normal	Possible	No

AKI: acute kidney injury; F: female; fT4_intox_; free thyroxine level at lithium intoxication; GFR: glomerular filtration rate; Li_max_: maximum lithium concentration at intoxication; TSH_intox_: thyroid-stimulating hormone at lithium intoxication; FT4I; free thyroxine index. ^a^ Calculated from creatinine, age and sex according to CKD-EPI formula. ^b^ Interpretation of the authors of the current study based on the information available in the original case reports. ^c^ Free thyroxine index = total thyroxine/thyroxine binding index (Thyroxine uptake ratio).

## Data Availability

The datasets generated and/or analysed during the current study are not publicly available due to a lack of ethics committee permission and not having been part of the consent process. The structure of the dataset and the coding specification are available from the authors. Any other reasonable request will be raised with the Swedish Ethical Review Authority and the healthcare provider.

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
