# Peer review of "Elevated Thyroxine Concentration and Lithium Intoxication—An Analysis Based on the LiSIE Retrospective Cohort Study"

_jcm, 2022, doi:10.3390/jcm11113041_

Round 1

Reviewer 1 Report

The authors have two more published papers on the LiSIE research program, one examining the correlation between lithium and hypothyroidism, the other transcribing paternal thyroid hormones in patients with bipolar or schizophrenic disorder, which means these are completely new hypotheses. which are being tested and which have not been tested before, certainly not under the LiSIE program in Sweden.

The disadvantage of the study is the very nature of the retrospective cohort as a methodology (were patients monitored for lithium exposure from 1997 to 2017? That was a long time ago, information can be unreliable, even lithium treatment standards may be different in 1997 and 2015) .  On the other hand, the authors cited this moment as a potential shortcoming of the study with additional emphasis on the fact that this was an observational study and that the second hypothesis was tested exclusively by clinical evaluation and not by monitoring tubular function during lithium exposure. 

All in all,  the article was written very well  The authors summarized their hypotheses, incidents in episodes by person-years seem real, briefly describe each of the previous studies on a similar topic ... Except for a few grammatical and format errors on page 7, I would not single out anything further.

I can recommend this article for publication.

Reviewer 2 Report

This is a well-written paper on a complex topic. Lieber et al. reported whether hyperthyroidism affects tubular renal function and results in lithium intoxication. Not much is known about these situations; therefore, additional information and cohort studies are welcomed to characterize these entities better. I have a few remarks and suggestions that I believe may improve the quality of the paper.

1) Page 4, in the materials and methods section, lines 114-121. Do you have information on FT3 levels? Based on thyroid autoantibodies and echocardiographic evaluation, how have you ruled out painless thyroiditis or autoimmune thyroiditis? The exposure time of hypothyroxinemia likely varies depending on the disease-causing hyperthyroidism.

2) Page 6, in the results section. It is interesting to note that symptoms such as Hand tremors, and diarrhea, common in hyperthyroidism, are similar in lithium addiction. The authors should also mention this point in the result.

3) Page 7, in the results section. Can you ascertain what percentage of patients had suspected nephrogenic diabetes insipidus due to lithium administration? I think that is another point that clinicians would like to know.

4) Page 9, in the discussion, line 303 and onward. As the authors also note, the limitation in this case series is that the impact of the onset of AKI and the addition of diuretics immediately before the onset of lithium intoxication is very significant. Although I do not think it affects the overall argument, I would have preferred to see a study that considered these points.
